# Construction of a Semantic Segmentation Network for the Overhead Catenary System Point Cloud Based on Multi-Scale Feature Fusion

**Tao Xu [1], Xianjun Gao [1], Yuanwei Yang [1,2,*], Lei Xu [3], Jie Xu [4] and Yanjun Wang [2,5]**

[1] School of Geosciences, Yangtze University, Wuhan 430100, China; 2021720578@yangtzeu.edu.cn (T.X.); junxgao@yangtzeu.edu.cn (X.G.)

[2] Hunan Provincial Key Laboratory of Geo-Information Engineering in Surveying, Mapping and Remote Sensing, Hunan University of Science and Technology, Xiangtan 411201, China; wangyanjun@hnust.edu.cn

[3] China Railway Design Corporation, Tianjin 300308, China; 2013102130022@whu.edu.cn

[4] National Petroleum and Natural Gas Pipeline Network Group Co., Ltd., Science and Technology Research Institute Branch, Langfang 065000, China; xujie02@pipechina.com.cn

[5] National-Local Joint Engineering Laboratory of Geo-Spatial Information Technology, Hunan University of Science and Technology, Xiangtan 411201, China

* Correspondence: 516042@yangtzeu.edu.cn; Tel.: +86-156-2354-2326

**Abstract:** Accurate semantic segmentation results of the overhead catenary system (OCS) are significant for OCS component extraction and geometric parameter detection. Actually, the scenes of OCS are complex, and the density of point cloud data obtained through Light Detection and Ranging (LiDAR) scanning is uneven due to the character difference of OCS components. However, due to the inconsistent component points, it is challenging to complete better semantic segmentation of the OCS point cloud with the existing deep learning methods. Therefore, this paper proposes a point cloud multi-scale feature fusion refinement structure neural network (PMFR-Net) for semantic segmentation of the OCS point cloud. The PMFR-Net includes a prediction module and a refinement module. The innovations of the prediction module include the double efficient channel attention module (DECA) and the serial hybrid domain attention (SHDA) structure. The point cloud refinement module (PCRM) is used as the refinement module of the network. DECA focuses on detail features; SHDA strengthens the connection of contextual semantic information; PCRM further refines the segmentation results of the prediction module. In addition, this paper created and released a new dataset of the OCS point cloud. Based on this dataset, the overall accuracy (OA), F1-score, and mean intersection over union (MIoU) of PMFR-Net reached 95.77%, 93.24%, and 87.62%, respectively. Compared with four state-of-the-art (SOTA) point cloud deep learning methods, the comparative experimental results showed that PMFR-Net achieved the highest accuracy and the shortest training time. At the same time, PMFR-Net segmentation performance on S3DIS public dataset is better than the other four SOTA segmentation methods. In addition, the effectiveness of DECA, SHDA structure, and PCRM was verified in the ablation experiment. The experimental results show that this network could be applied to practical applications.

**Keywords:** overhead catenary system point cloud; point cloud semantic segmentation; attention mechanism; multi-scale feature fusion; OCS point cloud dataset

## 1. Introduction

As an emerging mode of transportation, high-speed railways have been favored by more and more people because of their speed, stability, and high safety [1–3]. The OCS is an essential part of the railway system and plays a critical role in the train power supply. Still, influenced by uncertain factors, such as birds, strong winds, heavy rain, etc., the components of the OCS could become loose and even damaged. Consequently, the geometric parameters of the OCS would change, directly leading to an accident during the

operation of the train [4–6]. Therefore, to ensure the safety and reliability of the railway system, regular maintenance of the OCS is essential [7]. However, the traditional OCS geometric parameters are detected with manual portable measuring equipment along the track. This method relies on high human and time costs, and manual detection errors frequently occur [8]. Many researchers explored more intelligent detection methods to improve work efficiency and reduce cost. Installing cameras or Light Detection and Ranging (LiDAR) systems on the railway mobile transportation device to collect the OCS information is one of the mainstream methods. However, the photographs obtained by cameras hardly contain all the spatial information of the OCS, which are unable to restore the whole actual scene. Comparatively, the point cloud data of the OCS scanned by LiDAR reflect the spatial information in the actual scene more accurately [9,10]. These data help classify the OCS components for measuring geometric parameters effectively. Therefore, based on the point cloud data, it is of great significance to study the automatic segmentation of the OCS point cloud.

The automatic segmentation methods for point cloud data can be divided into two categories: traditional rules-based segmentation methods and statistical-based methods. Traditional rule-based segmentation principles include edge-based [11], region growing, model fitting [12], unsupervised clustering-based, etc. These methods are based on strict manual-designed features from geometric constraints, and they do not require supervised prior knowledge. However, they also have disadvantages. For example, the region-growing method is computation-intensive and requires a trade-off between accuracy and efficiency [13]. In terms of the statistics-based methods, machine learning methods are the most representative ones. "Regular" machine learning methods based on supervised classification operate on manually selected features with different classifiers, such as support vector machines (SVM) [14], random forest (RF) [15], maximum likelihood classifier (MLC) [16], and conditional random fields of statistical context models (CRF) [17,18], associative and non-associative Markov networks [19], etc. Prior knowledge is required for the initial sample and feature designs. The classification results are affected by the performance of the classifiers and the features. Recently, deep learning, also an important part of machine learning, has attracted the attention of researchers for its capability to obtain and combine multi-level features. Deep learning methods for point cloud segmentation mainly include multi-view-based [20], voxel-based [21], and raw-point-cloud-based methods. The multi-view-based method utilizes a dimensionality reduction strategy by converting 3D point cloud data to multi-view 2D images. The results are next restored to 3D after processing. Multi-view convolutional neural network (MVCNN) [22] is one of the most influential methods for multi-view [20]. However, the conversion between 2D and 3D leads to a loss of semantic features. Voxel-based points are voxelized and then processed by 3D convolution, which solves both unordered and unstructured problems of the raw point cloud. Voxelized data can be further processed by 3D convolution. Voxelization causes a certain loss of contour information. By increasing the resolution of voxels, the lost information can be reduced. However, the memory requirement would increase cubically [23].

Given the drawbacks of the aforementioned methods, many scholars have carried out further work based on directly processing raw point cloud data. PointNet [24] was the first deep learning network to process point cloud data directly. It could extract the semantic feature information of objects and solve the point cloud sorting invariance through multi-layer perceptron (MLP) and max-pooling. However, local features were always ignored, resulting in low prediction accuracy in some complex scenarios. PointNet++ [25], an upgraded version of PointNet [24], could enhance local feature extraction through sampling and grouping methods. In addition, the whole network structure was converted into the encoder–decoder style, using skipping connections to fuse the features extracted in the encoding and decoding stages. It could enhance the communication of contextual information. However, PointNet++ had a poor effect on the sparse point cloud. As a result, some linear components of the OCS, for example, the catenary wire and the contact wire, could not be detected effectively. A recently proposed network, MFF-A [26],

placed pyramidal pooling modules at the end of the network as an optimized structure for prediction. In this module, pooling windows of different sizes were used to extract multi-scale features, which improved the accuracy to a certain extent. Excessive semantic information could not be sensed yet due to continuous pooling operation. Further, an edge convolution (Edge Conv) structure was proposed by the dynamic graph convolutional neural network (DGCNN) [27]. This structure extracted local features by constructing the local domain of each point, computing edge features with the graph structure and assigning new features to its center points. However, due to the K-nearest neighbors (KNN) algorithm, the computing power requirements for the computer have increased. The PointSIFT [28] network used the encoder–decoder structure and introduced the scale invariant feature transform (SIFT) operator. Consequently, feature information could be described in eight directions to strengthen the extraction of local features. However, the training time of this network was extended due to the SIFT algorithm, resulting in low efficiency. A brand new convolution operation was proposed in the KPConv [29] network, which determined multiple core points within the sphere center range by taking one point as the sphere center. Further, the core points had their weight matrix to calculate the features of other points within the sphere and, finally, all features were fused as the features of the center point. The network used different matrices to update point features according to the various positions of each point. However, the method divided the whole point cloud into small point cloud blocks and then trained the small point cloud blocks, which affected the geometric feature extraction of large objects. As for the OCS point cloud segmentation, Lin et al. [5] proposed an OCS point cloud dataset. The original data of this dataset were collected by the SICK LiDAR system and artificially labeled about 16 km of point cloud data. They proposed a deep learning method for semantic segmentation of small scenes. This method had high requirements for data preprocessing and needed to extract the scene in a single frame. Chen et al. [6] also proposed their own catenary point cloud dataset. This dataset divided the data into two groups: 50 m and 100 m for experiments. They proposed a clustering point cloud classification method, which could extract the contact wire and catenary wire with high accuracy. However, this method was only for linear OCS components and could not complete the extraction task of other OCS components. The two datasets of the OCS point cloud were not publicly published.

This paper proposes a point multi-scale feature fusion refinement network (PMFR-Net) to achieve effective semantic segmentation of the OCS point cloud. Compared with other networks, the proposed PMFR-Net includes double efficient channel attention (DECA), serial hybrid domain attention (SHDA) structure, and point cloud refinement module (PCRM) for multi-scale feature extraction and fusion. Based on efficient channel attention (ECA) [30], DECA has made an adaptive improvement for point cloud data, which can effectively filter complex semantic information and improve the representation ability of the model. In addition, the features obtained in the encoding stage were filtered through the SHDA before being fused with the features extracted in the decoding stage, strengthening the connection of contextual semantic information. At the end of the prediction module, an improved PCRM based on atrous spatial pyramid pooling (ASPP) [31] was connected. The PCRM extracts multi-scale features through multi-layer dilated convolution with different dilation rates. Moreover, the model fuses the feature maps of each level in turn, so that the semantic features of the point cloud can be better integrated. In this study, under the premise of ensuring the robustness of the network, the network complexity is controlled so that it can achieve high accuracy.

The main contributions of this paper are summarized as follows:

1.  In this paper, the PMFR-Net is proposed for semantic segmentation of the OCS point cloud. The network integrates the DECA, SHDA, and PCRM. PMFR-Net is superior to the other four SOTA methods in visual discrimination and quantitative evaluation.
2.  A point cloud refinement module (PCRM) is designed to achieve multi-scale feature extraction through multi-layer dilated convolution channels with different dilation rates. A multi-level feature fusion structure is created. This module is beneficial

for extracting the semantic features of the same component at different scales and strengthens the extraction of local features.

3. A new manual-labeled OCS point cloud dataset is established in this paper. This dataset contains 88 separate scenes, including two kinds of structure: single-arm and double-arm scenarios. The point cloud is subdivided into nine categories for labeling.

The rest of the paper is organized as follows: Section 2 discusses the implementation details of PMFR-Net. Section 3 presents the OCS point cloud dataset, experimental environment, and comparative experiments. Section 4 demonstrates the effectiveness of each innovative module in the network. The significance of this paper is summarized in Section 5.

## 2. Methods

This section mainly introduces the method proposed in this paper. First, the overall framework of PMFR-Net is outlined in Section 2.1. Second, the prediction module is presented in Section 2.2. Then, the SHDA structure and DECA structure are introduced in Sections 2.2.1 and 2.2.2, respectively. The PCRM proposed in this paper is introduced in the last section.

### 2.1. Overall Framework

In order to better solve the problems of the low accuracy and efficiency of the OCS point cloud segmentation, this paper proposes a deep learning neural network, PMFR-Net, as shown in Figure 1.

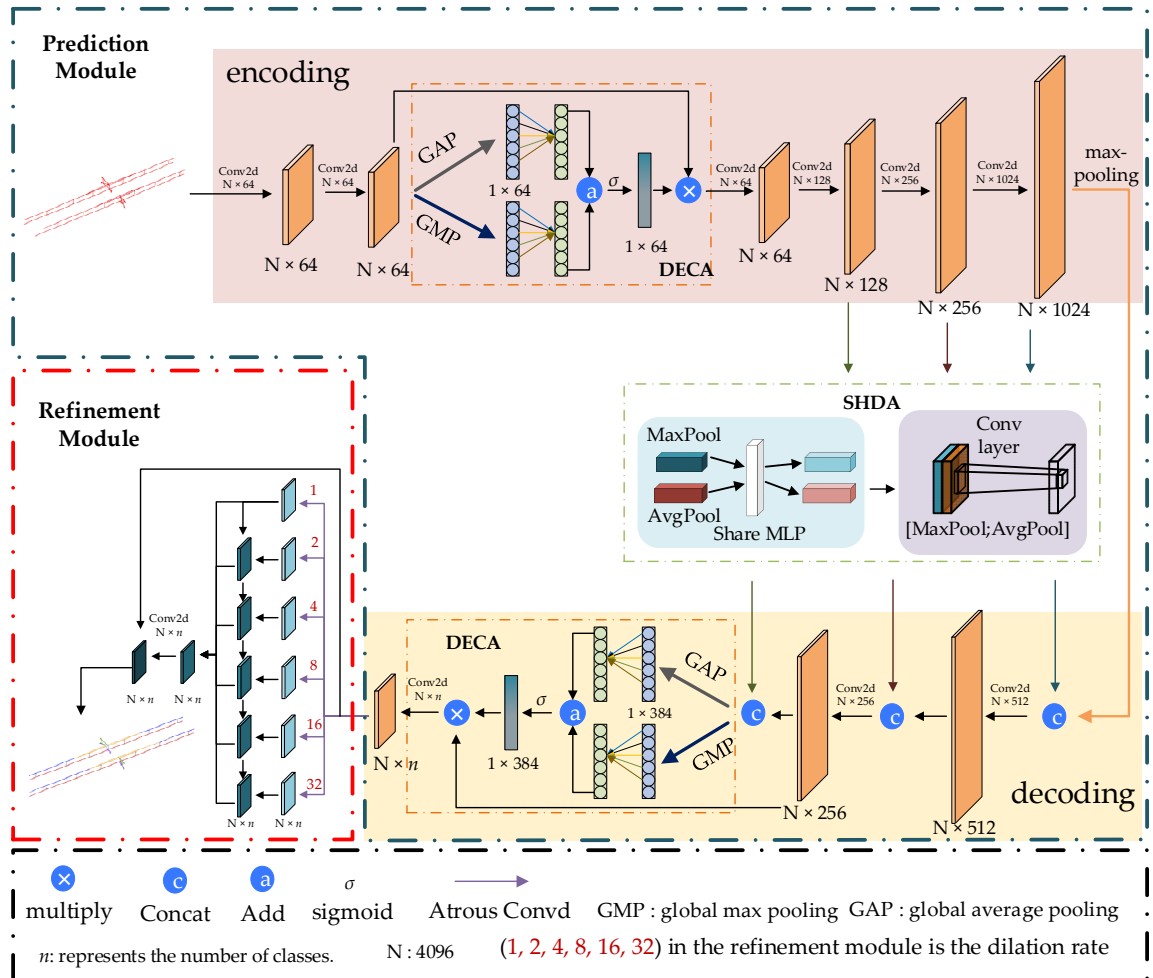

**Figure 1.** Overall framework diagram of the proposed PMFR-Net.

The PMFR-Net network is divided into the prediction module and the refinement module. The prediction module drives the overall network by the convolution of multiple shared weights to extract the point cloud features, denoted as PointSimpleNet. On this basis, the DECA module is inserted twice in the encoding and decoding stages, respectively, to realize cross-channel interaction and improve the representation ability of the network. The features extracted by the convolution using N × 128, N × 256, and N × 1024 in the encoding stage are filtered through the SHDA structure and then fused with the features obtained by N × 256, N × 512, and max-pooling in the decoding stage. It strengthens the communication of context semantic information. Then, the refinement module uses the PCRM to further optimize the feature map obtained by the prediction module. This module aggregates multi-scale semantic information through dilated convolution to enhance the grasp of overall and detailed features. The network realizes high-precision semantic segmentation for the OCS point cloud.

### 2.2. The Prediction Module

The prediction module extracts semantic information of point cloud by convolution operation, and the SHDA structure and DECA mechanism are added. The SHDA structure strengthens the integration of contextual semantic information, and the DECA mechanism can boost the communication and fusion of semantic information in a cross-channel manner.

#### 2.2.1. The Serial Hybrid Domain Attention Structure

Classical networks with skip connection structures include U-Net [32] and a series of networks based on U-Net. This paper uses a serial hybrid domain attention structure based on skip connections. In this structure, the features of the encoding stage are filtered by the SHDA to retain the geometric information and reduce interference information. Then they are integrated into the features in the decoding stage. The addition of geometric features can effectively deal with the boundary of catenary components, thereby improving the segmentation accuracy of the overall network.

Based on the CBAM [33], the attention module is driven bi-directionally through the channel and spatial attention models. It can effectively filter the interference information and highlight the geometric information features. The channel attention module compresses the spatial dimension of the feature map through max-pooling and average-pooling, respectively, and then inputs the obtained results into the spatial attention mechanism. Average-pooling and max-pooling are performed in the channel dimension in the spatial attention module. Then the two pooling results are superimposed together, and finally, the convolution operation is used on the spliced feature map to generate the final spatial attention feature map. It can effectively filter the interference information and highlight the geometric information features. Equation (1) shows the channel attention module, as shown in the blue box in Figure 2. The spatial attention module is shown in Equation (2), as shown in the purple box in Figure 2.

$$M_c(F) = \sigma(\text{MLP}(\text{MaxPool}(F)) + \text{MLP}(\text{AvgPool}(F))), \tag{1}$$

where $F$ represents the feature map, $\sigma$ is the sigmoid function, $M_c(F)$ represents the features obtained by the channel attention module, MLP represents the multi-layer perceptron operation, MaxPool represents max-pooling, and Avgpool represents average-pooling.

$$M_s(F) = \sigma\left(f^{7\times7}([\text{AvgPool}(F); \text{MaxPool}(F)])\right), \tag{2}$$

where $\sigma$ is the sigmoid function, $f$ represents the convolution, $7 \times 7$ represents the size of the convolution kernel, $F$ represents the feature map, $M_s(F)$ represents the features obtained by the spatial attention module. MaxPool represents max-pooling, Avgpool represents average-pooling, and [AvgPool($F$); MaxPool($F$)] represents the result of average-pooling superimposed on the result of max-pooling.

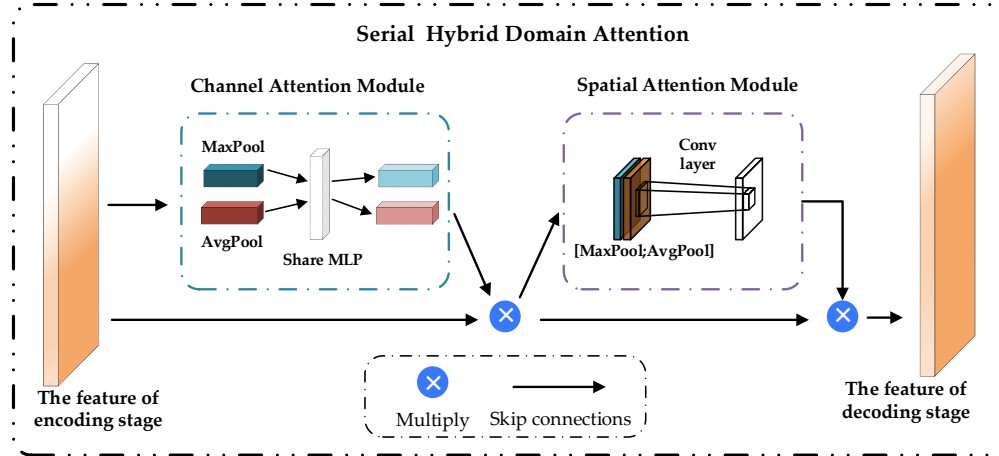

**Figure 2.** Structure diagram of the serial hybrid domain attention (SHDA).

2.2.2. Double Efficient Channel Attention

In recent years, attention mechanism has been widely applied in various networks by researchers, among which the classic ones are SE-Net [34], ECA-Net [30], and DA-Net [35]. This paper designs an improvement on ECA-Net, denoted as the DECA module, as shown in Figure 3. This module can aggregate the convolution features using global average-pooling and global max-pooling without dimensionality reduction and perform local cross-channel interactions. As shown in Equation (3), *K* represents the coverage of local cross-channel exchange. Channel interaction is realized through conv1d, as shown in Equation (4). The result F$_{DECA}$ in DECA can be calculated using Equation (5).

$$K = \left| \frac{\log_2(C)}{\gamma} + \frac{b}{\gamma} \right|_{odd}, \tag{3}$$

where $|t|_{odd}$ is the nearest *odd* number to *t* and *C* is the number of channels, $\gamma = 2$, $b = 1$, respectively.

$$\omega_i = \sigma \left( \sum_{j=1}^{K} w_i^j y_i^j \right), y_i^j \in \Omega_i^K \tag{4}$$

where $\omega_i$ is the result of channel interaction, $w_i^j$ the represented the weight of the channel feature, $y_i$ denotes adjacent characteristic channel in one-dimensional space. *K* is the result calculated by Equation (3), *i* represents the number of channels, and $j \in K$, $\sigma$ is the sigmoid activation function.

$$F_{DECA} = F \bigotimes \sigma(\text{Conv1d}(\text{GMP}(F)) + \text{Conv1d}(\text{GAP}(F))) \tag{5}$$

where *F* represents the input feature map, $\sigma$ is the sigmoid activation function, $\otimes$ is the element-wise product, GAP stands for global average-pooling, GMP stands for global max-pooling, and conv1d represents one-dimensional convolution.

Global average-pooling (GAP) and global max-pooling (GMP) are used to aggregate semantic features simultaneously. Global average-pooling is suitable for extracting the overall global features. Still, the grasp of details is slightly lacking, so a layer of global max-pooling is performed in parallel to enhance the ability to detect details. For example, the aggregation of information at the connections between different OCS components can improve segmentation accuracy. These two pooling methods complement each other and play a positive guiding role in the overall model information extraction ability, significantly strengthening the extraction of local semantic features.

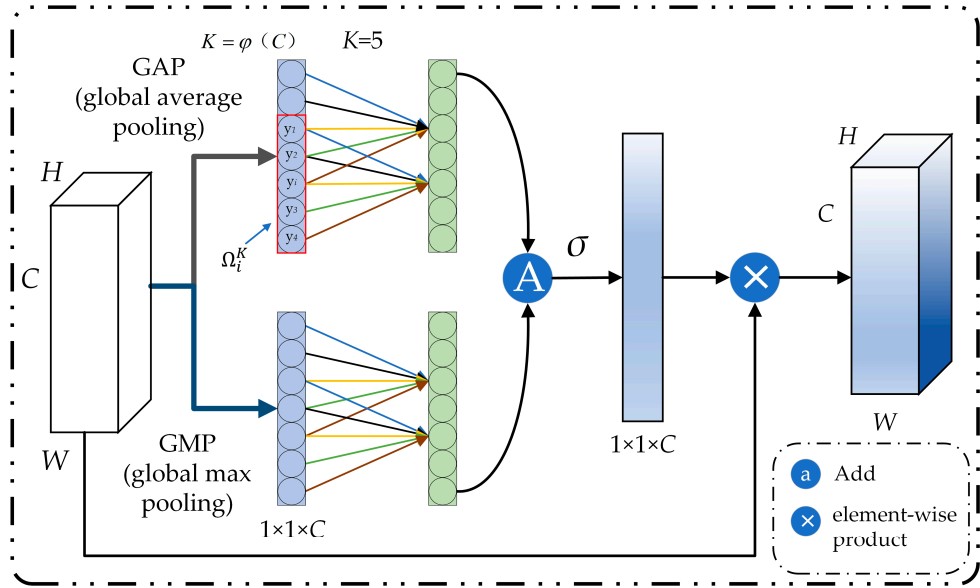

**Figure 3.** Double efficient channel attention (DECA) module diagram. GAP stands for global average-pooling and GMP stands for global max-pooling.

### 2.3. Point Cloud Refinement Module

The extraction and fusion of multi-scale features are essential in complex scenes, considering the feature extraction of objects of different sizes. The OCS structure comprises multiple components with various features, and the scene is more complicated. To further improve the representation ability of the model, this paper proposed a point cloud refinement module called PCRM, as shown in Figure 4. PCRM uses a dilated convolution structure and is located at the end of the prediction module network. Different receptive fields are obtained through dilated convolution with varying dilation rates, which improves the accuracy of the large segmentation objects. The elastic catenary wire, contact wire, and catenary wire of the OCS point cloud data are relatively larger than other parts. In the case of the small receptive domain, it is difficult to perceive the whole object, which will lead to the incomplete extraction of object features and low precision of semantic segmentation. Thus, multi-layer dilated convolution is used to obtain relatively complete overall features at different scales. Through $n$ parallel convolution operations with varying dilation rates, $n$ feature maps $CF_i$ ($i \leq n$) of different scales are obtained. The feature maps of each level are fused in turn and activated by convolution and then merged into the initial input result of the module to get the refined result. The different layers are combined in sequence to obtain the fused feature map $MF_j$ ($j < n$), defined in Equation (6). Then the fused feature map $MF_j$ and the feature map $CF_i$ with a total of $n$ feature maps are fused to obtain $O_F$ according to Equation (7). Finally, the result of a convolution operation and the result obtained by the prediction module are fused to get the final segmentation result $O_{sem}$, as shown in Equation (8).

$$MF_j = CF_{i-1} + CF_i, \ (1 < i \leq n), \tag{6}$$

$$O_F = \left( \sum_{j=1}^{n} MF_j \right) + CF_i, \tag{7}$$

$$O_{sem} = \text{Conv2d}(O_F) + F_{input}, \tag{8}$$

where $F_{input}$ is the feature map of the input refinement module, $CF_i$ ($i \leq n$) is the feature map of dilated convolution, $MF_j$ ($j < n$) is the fused feature map, and $n$ is the number of dilated convolutions.

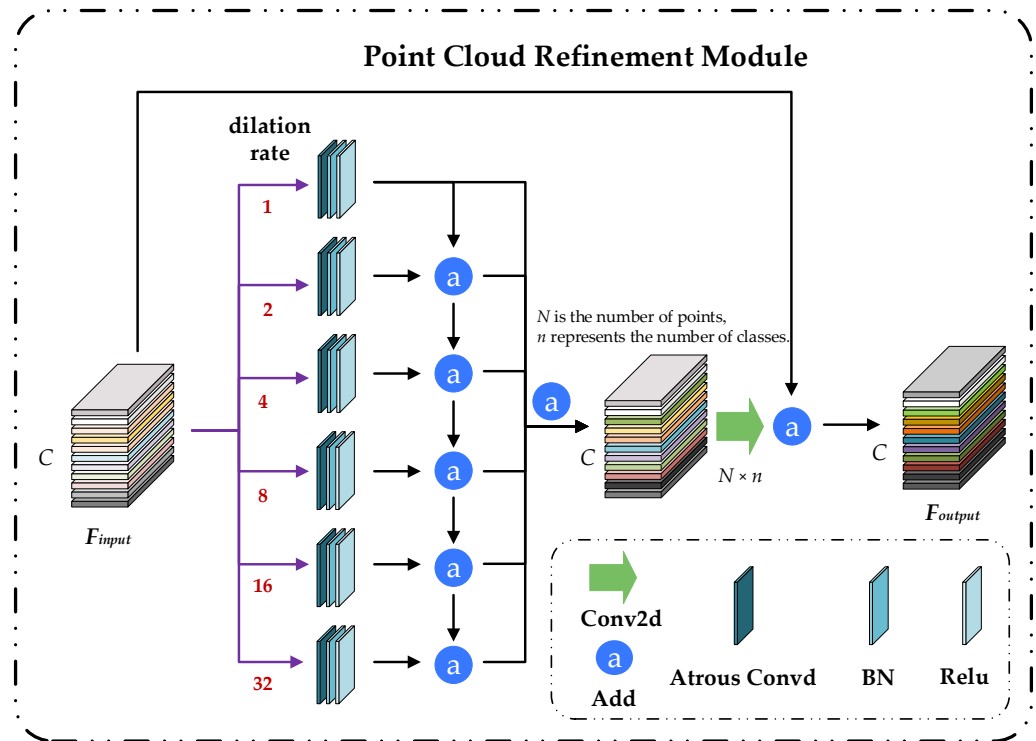

**Figure 4.** Point cloud refinement module (PCRM) diagram.

In Equation (6), *n* was 6, and *i* took 1. The structure diagram of the refinement module is shown in Figure 4. Different colors represent different channels, and the change in color depth shows the degree of feature information extraction. Compared with the previous simple series fusion, this method can be changed into layers and integrated with sequence, effectively reducing the omission of features and improving the semantic information of the OCS components as much as possible.

## 3. Experiment and Result Analysis

In this section, an experimental evaluation of the effectiveness of PMFR-Net is presented and compared with four other SOTA methods. Section 3.1 introduces the proposed OCS point cloud dataset. Section 3.2 describes experimental settings and evaluation indicators. Lastly, Section 3.3 presents experimental results, analysis, and comparison.

### 3.1. Experimental Data

In recent years, to promote the research on point cloud recognition, some large datasets have been established, such as S3DIS [36], ShapeNet [37], and ScanNet [38]. Most of the public point cloud datasets are for indoor scenes of buildings, individual objects, and outdoor scenes. Datasets for railway scenes are rare. Therefore, we built a new OCS point cloud dataset for our experiments by annotating semantic information manually and we will make the created OCS dataset open access for further public study.

The dataset is collected from the actual point cloud data of some sections of the high-speed railway from Nantong to Yancheng. The point cloud along the railway was scanned by LiDAR using Optech Lynx HS 600 VMMS. Each point covered seven attributes (*X*, *Y*, *Z*, *R*, *G*, *B*, *I*). Before making the dataset, the original high-speed railway point cloud data were preprocessed to remove the non-OCS point cloud. As shown in Figure 5, the red point marks the OCS data, which are extracted for subsequent manual labeling work. The entire OCS scene is about 4 km and it consists of 88 separate scenes. Each scene is 40–50 m long, nearly 10 m wide, and about 1.5 m high. The adjacent scenes are continuous. Each scene is subdivided into nine classes: catenary wire, steady arm, oblique cantilever, straight cantilever, elastic catenary wire, registration arm, others, dropper, and contact wire. The

88 separate scenes can be divided into single-arm and double-arm OCS scenes, as shown in Figure 6. The number of points in the entire dataset scene is around 50,000,000. Among them, the point number of the single-arm scene is about 520,000, and the point number of the double-arm scene is about 700,000. The detailed OCS dataset data information is shown in Table 1. There are 15 scenes in the validation set, 17% of the total dataset. The proportion of the single-arm and double-arm in the validation set is similar to that in the training set, which is almost 3:1. At the same time, three scenes are randomly selected in the dataset for testing, including one double-arm scene and two single-arm scenes. The dataset is be available at https://github.com/Waynexutao/DatasetAccess.git, (accessed on 4 May 2022) to assist other researchers in conducting relevant research.

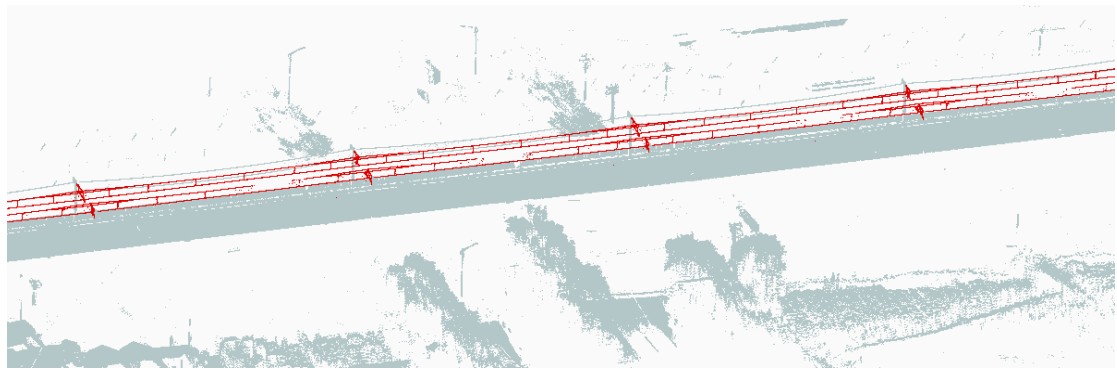

**Figure 5.** The OCS extraction point cloud result by preprocessing (red is the OCS extraction result, gray is the background point).

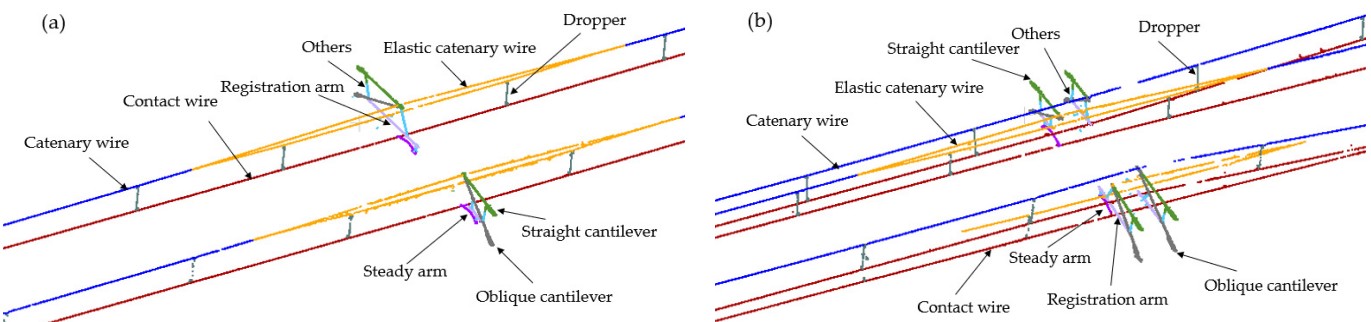

**Figure 6.** Component diagram of the OCS: (**a**) is the single-arm OCS and (**b**) is the double-arm OCS.

**Table 1.** Data distribution introduction in the OCS dataset.

| | Single-Arm | Double-Arm | Training | | Validation | | Testing | |
|---|---|---|---|---|---|---|---|---|
| | | | Single-Arm | Double-Arm | Single-Arm | Double-Arm | Single-Arm | Double-Arm |
| The number of scenes | 68 | 20 | 55 | 15 | 11 | 4 | 2 | 1 |
| The number of points | $3.54 \times 10^7$ | $1.4 \times 10^7$ | $2.9 \times 10^7$ | $1.1 \times 10^7$ | $5.7 \times 10^6$ | $2.8 \times 10^6$ | $1.04 \times 10^6$ | $7 \times 10^5$ |

Except for the proposed OCS dataset in this paper, the S3DIS dataset [36] is also tested to validate the performance of the proposed network. S3DIS dataset is a multi-region indoor point cloud dataset covering six large-scale indoor areas and 272 rooms. The point number is over 215 million points and the area is over 6000 m². It is an RGB-D point cloud dataset with pixel-level semantic markers. The point cloud can be divided into 13 classes: ceiling, floor, wall, beam, column, window, door, table, chair, sofa, bookcase, board, and sundries. The data distribution of the S3DIS is shown in Table 2. The whole Area 6 with 48 rooms is used as the validation set in our experiment. Among them, seven different

types of rooms are randomly selected as the test set in the whole dataset. The S3DIS dataset is shown in Figure 7.

**Table 2.** Distribution of the number of rooms for the S3DIS dataset.

| | Area 1 | Area 2 | Area 3 | Area 4 | Area 5 | Area 6 | Total | | |
|---|---|---|---|---|---|---|---|---|---|
| | | | | | | | Training | Validation | Testing |
| The number of rooms | 44 | 40 | 23 | 49 | 68 | 48 | 217 | 48 | 7 |

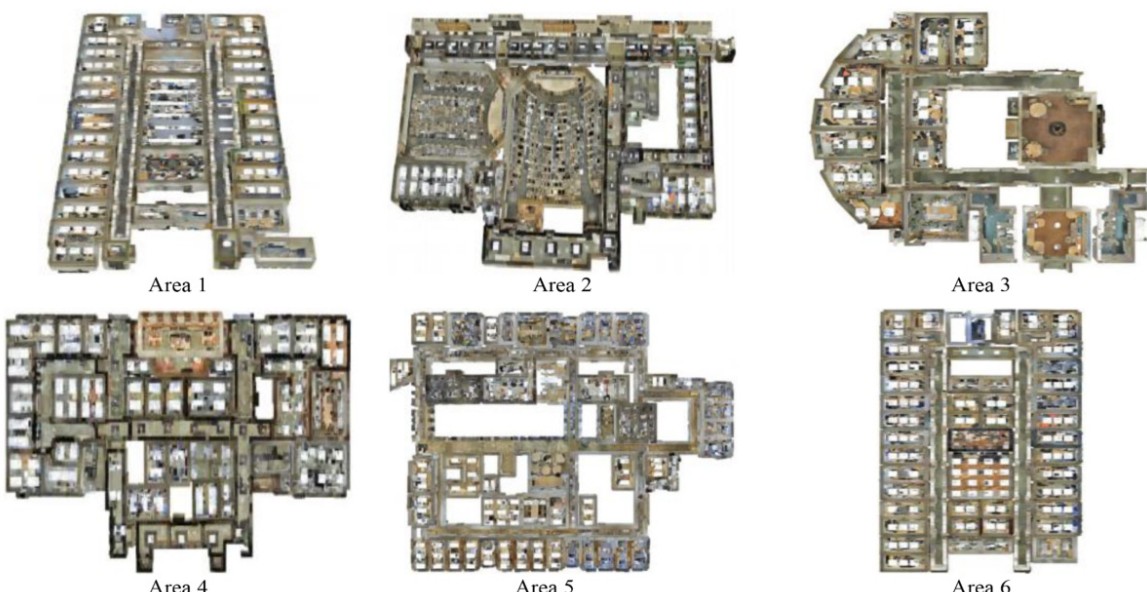

**Figure 7.** The S3DIS dataset contains thumbnails of Area 1 to Area 6.

The similarity between these data and the OCS dataset proposed in this paper are:

1. Both datasets contain multiple objects of different scales. For example, walls and ceilings in the S3DIS dataset belong to large-scale categories, while tables, chairs, etc., belong to small-scale categories. The OCS dataset has the most obvious difference in scale between conductors, tension cables, and hanging strings.
2. Both datasets contain objects with similar geometric features. For example, the walls, ceilings, and floors in the S3DIS dataset are all dense surface features, while the wires and bearing cables in the OCS dataset are similar linear features.

### 3.2. Experimental Settings and Evaluation Indicators

#### 3.2.1. Experimental Settings

All experiments in this article were conducted on Windows 10 (64-bit) workstations. The workstation configuration is Intel(R) Core (TM) i7-9700 KCPU @ 3.60 GHz, 32 GB memory, and a GPU of NVIDIA GeForce RTX 2080Ti with an 11 GB RAM. All networks were implemented on TensorFlow 1.14 and Keras 2.2.4.

In our experiment, every point is represented by an 8D vector $\mathbf{P}$ = {$X$, $Y$, $Z$, $R$, $G$, $B$, $I$, $T$}, where $X$, $Y$, and $Z$ represent 3D coordinates; $R$, $G$, and $B$ represent the color information; $I$ means the intensity information; $T$ denotes the class value. Because the intensity information difference between each class is not obvious, it is not used in this paper. We used the information of ($X$, $Y$, $Z$, $R$, $G$, $B$, $T$) for the experiment. The training and verification samples of the network are hierarchical data format (HDF5) point cloud data and the num point was 4096. The batch size of HDF5 was 1000. In the training stage, the batch size was 12, and the training epoch was set to 100. The learning rate was dynamic: the initial learning rate was 0.0001 and the lowest was 0.00001 at the end of training.

The proposed network adopts the Cross-Entropy Loss function [39], shown in Equation (9), as the loss function.

$$\text{Loss} = -\frac{1}{N} \sum_{i=1}^{N} \sum_{c=1}^{M} y_{ic} log_2(p_{ic}), \tag{9}$$

where $N$ is the number of samples and $M$ is the number of classes. $y_{ic}$ is the label value when the true class of sample $i$ is equal to $c$, $y_{ic}$ takes 1, otherwise it is 0. $p_{ic}$ is the predicted probability that sample $i$ belongs to class $c$.

### 3.2.2. Evaluation Indicators

Five evaluation indexes commonly used in semantic segmentation evaluation were selected to evaluate the performance. They are overall accuracy (OA), precision (P), recall rate (R), F1-Score, and mean intersection over union (MIoU). OA refers to the proportion of all points correctly classified to all points involved in evaluation calculation, and its calculation equation is shown in Equation (10). The precision refers to the proportion of positive samples correctly predicted among all the results predicted as positive, as shown in Equation (11). The recall refers to the proportion of positive samples with correct prediction in all positive samples, as shown in Equation (2). F1-Score refers to the harmonic average of accuracy and recall rate, which is a comprehensive evaluation index, as shown in Equation (13). The IoU is the intersection ratio of all predicted positive class points and real positive class points over their union. MIoU is the average value of IoU and its calculation equation is shown in Equation (14).

$$\text{OA} = \frac{\text{TP} + \text{TN}}{\text{TP} + \text{TN} + \text{FP} + \text{FN}}, \tag{10}$$

$$\text{P} = \frac{\text{TP}}{\text{TP} + \text{FP}}, \tag{11}$$

$$\text{Recall} = \frac{\text{TP}}{\text{TP} + \text{FN}}, \tag{12}$$

$$\text{F1} - \text{Score} = \frac{2 \times \text{P} \times \text{R}}{\text{P} + \text{R}}, \tag{13}$$

$$\text{MIoU} = \frac{\sum_{i=1}^{c} \text{IoU}_i}{c}, \tag{14}$$

where TP (true-positive) is the number of points that correctly identified positive samples. TN (true-negative) is the number of points that are actually negative samples but are identified as positive samples. FP (false-positive) is the number of points where negative samples are correctly identified. FN (false-negative) is the number of points that are actually positive samples but are identified as negative samples.

### 3.3. Experimental Results and Analysis

Section 3.3.1 shows and analyzes the quantitative and visual results of the semantic segmentation results obtained by the PMFR-Net on the OCS point cloud dataset. In Section 3.3.2, three typical semantic segmentation methods of point cloud and one method especially for the segmentation of OCS point cloud were selected for comparative experiments. In Section 3.3.3, the experimental results of our approach and other methods on the S3DIS public dataset will be compared. All the following accuracy evaluation metrics come from the test set. The test set of the OCS dataset consists of three scenes (including two single-arm scenes and one double-arm scene), and the S3DIS test set consists of seven randomly selected different rooms.

### 3.3.1. Segmentation Results of PMFR-Net

The quantitative evaluation results of prediction accuracy are shown in Table 3. After visualization, the predicted point cloud data are shown in Figure 8. As shown in Table 1, the

F1-Score of the contact wire is the highest, followed by the oblique cantilever and catenary wire, and the steady arm. The precision of these four components is more than 95%, since their spatial position and shape characteristics are apparent. For example, the contact wire and catenary wire are parallel to each other. There is a certain angle between the oblique cantilever and the straight cantilever. Therefore, a high segmentation accuracy can be achieved. From four different evaluation results, it can be concluded that the segmentation accuracy of the dropper is the worst. There may be two reasons for this phenomenon. First, the dropper is small, and it is easy for it to be incompletely scanned, which leads to unclear semantic features. As a result, it increases the difficulty of network learning. Second, because the point number of the dropper is much less than other parts, the proportion of the dropper to the whole OCS points is also tiny, which leads to an imbalance in semantic network segmentation training.

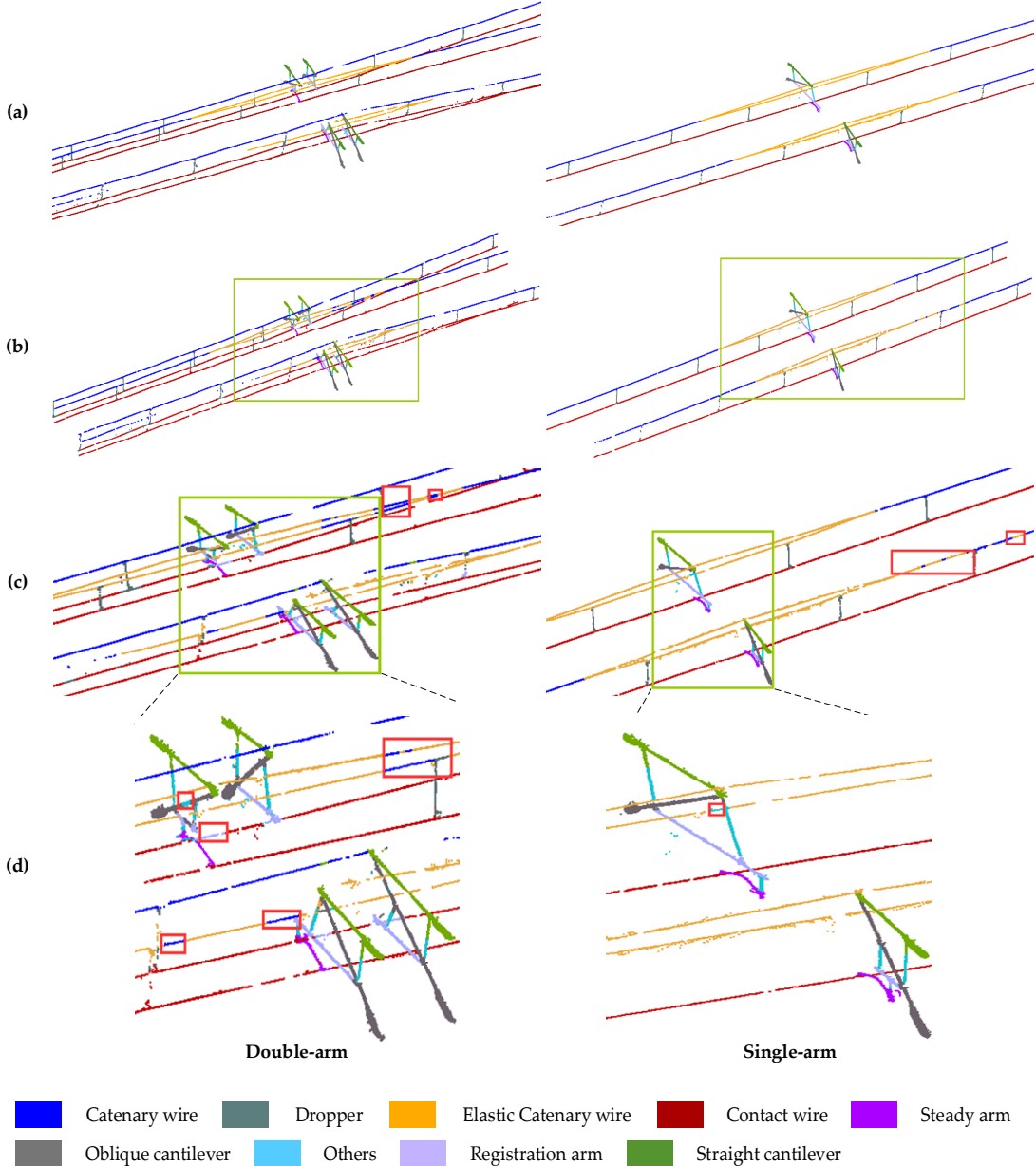

**Double-arm**                                                  **Single-arm**

■ Catenary wire ■ Dropper ■ Elastic Catenary wire ■ Contact wire ■ Steady arm
■ Oblique cantilever ■ Others ■ Registration arm ■ Straight cantilever

**Figure 8.** Prediction results of single-arm and double-arm. (**a**) is the true value, (**b**) is the semantic segmentation result of PMFR-Net, (**c**) is the content in the green box in (**b**), and the red box is the wrongly classified point cloud, (**d**) is the content in the green box in (**c**), and the red box is the wrongly classified point cloud.

**Table 3.** Precision evaluation comparison of the OCS dataset semantic segmentation. The overall accuracy evaluation of the two scenes is shown in bold.

| | OA (%) | | F1-Score (%) | | MIoU (%) | |
|---|---|---|---|---|---|---|
| | Single-Arm | Double-Arm | Single-Arm | Double-Arm | Single-Arm | Double-Arm |
| Catenary wire | 94.61 | 96.34 | 94.41 | 96.39 | 89.43 | 93.03 |
| Steady arm | 97.81 | 94.66 | 95.04 | 93.84 | 90.55 | 88.39 |
| Oblique cantilever | 98.87 | 96.65 | 99.06 | 92.15 | 98.14 | 85.45 |
| Straight cantilever | 98.02 | 81.59 | 98.01 | 89.55 | 96.10 | 81.08 |
| Elastic catenary wire | 92.92 | 91.36 | 93.19 | 89.40 | 87.26 | 80.84 |
| Registration arm | 96.52 | 88.16 | 93.74 | 91.61 | 88.21 | 84.51 |
| Dropper | 91.69 | 68.13 | 90.72 | 73.19 | 83.02 | 57.71 |
| Contact wire | 99.53 | 98.92 | 99.67 | 97.79 | 99.35 | 95.68 |
| average | **95.77** | | **93.24** | | **87.68** | |

The segmentation results in Figure 8 show that since the single-arm scene is relatively simple, the segmentation results are better. There is only a tiny mis-segmentation between the elastic catenary wire and catenary wire. In contrast, the double-arm scene is more complex. The segmentation result has certain misclassifications, mainly at the border of the elastic catenary wire and the catenary wire. However, other parts can be divided more thoroughly, such as between the oblique cantilever and the straight cantilever. To sum up, PMFR-Net has good segmentation ability for the OCS point cloud.

### 3.3.2. Comparative Experiment on the OCS Dataset

The comparative experiment selects the classical point cloud segmentation network PointNet [24], PointNet++ [25], DGCNN [27], and OCS segmentation network MFF-A [26] for comparative analysis with the proposed PMFR-Net, to further evaluate its effectiveness. We did not make any changes to the network of the four comparison methods. To ensure the fairness of the experiment, each group of experiments was set up and assessed following the experimental setting and evaluation standards in Section 3.2.

Table 4 lists all evaluation metrics of the five comparison methods tested on the OCS dataset. In contrast, the PMFR-Net proposed in this paper obtained the highest accuracy values. The OA, MIoU, and F1-Score of PMFR-Net are higher than the following best method, PointNet++ by a more noticeable improvement of about 1.6%, 2.5%, and 2.3%, respectively. Moreover, the training time of PMFR-Net is 0.5 hours faster than that of Pointnet++. In addition, as shown in Table 4, the accuracy of all methods for single-arm scenes is higher than that for double-arm scenes. However, regarding the performance difference in two segment scenes, PointNet++ shows the best balance ability, followed by the PMFR-Net. Moreover, the F1-Score difference between the singe-arm and double-arm scene of PointNet++ is only 0.5%, and PMFR-Net is about 5%. However, the accuracy of PMFR-Net in single-arm scenes is higher than that of PointNet++ and the performance difference between PMFR-Net and PointNet++ to segment double-arm scenes is tiny. Therefore, the overall semantic segmentation effect of PMFR-Net is the highest.

As shown in Figure 9, the proposed method is superior to other methods, indicating that PMFR-Net can effectively segment the scene of the OCS point cloud. PointNet obtained the lowest accuracy. It takes the feature map with N × 1024 channels as a local feature and the result of max-pooling as a global feature, which are fused to obtain a feature map with the most abundant feature information. However, the max-pooling causes a significant loss of features. Since the network represents each point through MLP, the ability to integrate local structural information and connect contextual semantic information is weak. There are many misclassifications in each category. PointNet++ used the encoder–decoder structure as a whole in the network and utilized skip connection to strengthen the context semantic information. However, the sparsity of the point cloud has a significant influence on the performance of PointNet++. The OCS point cloud is relatively sparse in linear components, so many misclassifications exist between the catenary wire and elastic catenary wire.

Concerning the DGCNN, the most prominent feature can be extracted through Edge Conv. The method constructs a K-nearest neighbors (K-NN) graph to represent the local features through the features of each point. Because the K-NN algorithm needs to traverse every point, the DGCNN requires high computing power. Regarding MFF-A, a pyramid pooling module is added at the end of the network to form a multi-scale feature extractor. However, the pooling operation caused a significant loss of semantic information, and it is challenging to play a good role in optimization and refinement. In contrast, in the prediction module of PMFR-Net, the features of the encoding stage are filtered through the SHDA in advance and then integrated into the features of the decoding stage. This strategy enhanced the context semantic information aggregation. The refinement module adopts the dilated convolution pyramid to extract and fuse multi-scale features and optimizes the results of the prediction module. Due to the combination of these two mechanisms, PMFR-Net can effectively classify the OCS point cloud, especially the simple single-arm scene. The segmentation result is more precise.

**Table 4.** Quantitative evaluation of the comparative methods on the OCS point cloud dataset (%). The average is underlined and the best metrics are highlighted in bold.

| Method | Scene_Class | OA | Precision | Recall | MIoU | F1-Score | Time(h) |
|---|---|---|---|---|---|---|---|
| PointNet [24] | single-arm | 86.24 | 86.53 | 79.23 | 71.06 | 82.06 | |
| | double-arm | 88.50 | 75.79 | 82.46 | 65.37 | 77.74 | 5 |
| | average | <u>87.37</u> | <u>81.16</u> | <u>80.84</u> | <u>68.22</u> | <u>79.90</u> | |
| PointNet++ [25] | single-arm | 95.90 | 91.61 | 91.06 | 86.17 | 91.12 | |
| | double-arm | 93.88 | 90.31 | 91.35 | 83.23 | 90.62 | 4.5 |
| | average | <u>94.11</u> | <u>91.18</u> | <u>91.16</u> | <u>85.19</u> | <u>90.97</u> | |
| DGCNN [27] | single-arm | 93.71 | 94.23 | 90.23 | 85.64 | 92.11 | |
| | double-arm | 92.40 | 88.07 | 86.45 | 77.56 | 86.61 | 5 |
| | average | <u>93.18</u> | <u>91.76</u> | <u>88.53</u> | <u>82.13</u> | <u>89.89</u> | |
| MFF-A [26] | single-arm | 95.89 | 94.93 | 94.25 | 89.79 | 94.54 | |
| | double-arm | 92.50 | 86.24 | 87.87 | 77.87 | 86.73 | 5.5 |
| | average | <u>94.53</u> | <u>90.59</u> | <u>91.06</u> | <u>83.83</u> | <u>90.64</u> | |
| Ours | single-arm | 96.58 | 96.25 | 94.76 | 91.51 | 95.48 | |
| | double-arm | 94.57 | 89.48 | 91.92 | 83.34 | 90.49 | **4** |
| | average | <u>**95.77**</u> | <u>**92.97**</u> | <u>**93.54**</u> | <u>**87.62**</u> | <u>**93.24**</u> | |

### 3.3.3. Comparative Experiment on the S3DIS Dataset

This paper also uses the public dataset S3DIS for experiments to verify the generalization of the model. Meanwhile, the four methods, including PointNet, PointNet++, DGCNN, and MFF-A, are also compared with the proposed method. Area_1–Area_5 in the S3DIS dataset are all selected as the training set data, and Area_6 is selected as the test set. Due to a large amount of overall data, the epoch of PointNet++ is set to 32 and the epoch of other methods is set to 100.

From comparing the accuracy in Table 5, the PMFR-Net got the highest among the five accuracy evaluation indicators. Compared with PointNet, MIoU is increased by nearly 20%, F1-Score value is increased by nearly 15%, and OA is increased by about 5%. Although PMFR-Net takes 8 hours to train for 100 epochs, it is not the least amount of time, but it is only slightly higher than PointNet. Thus, it can be concluded that PMFR-Net has a specific generalization and can achieve good accuracy in different datasets.

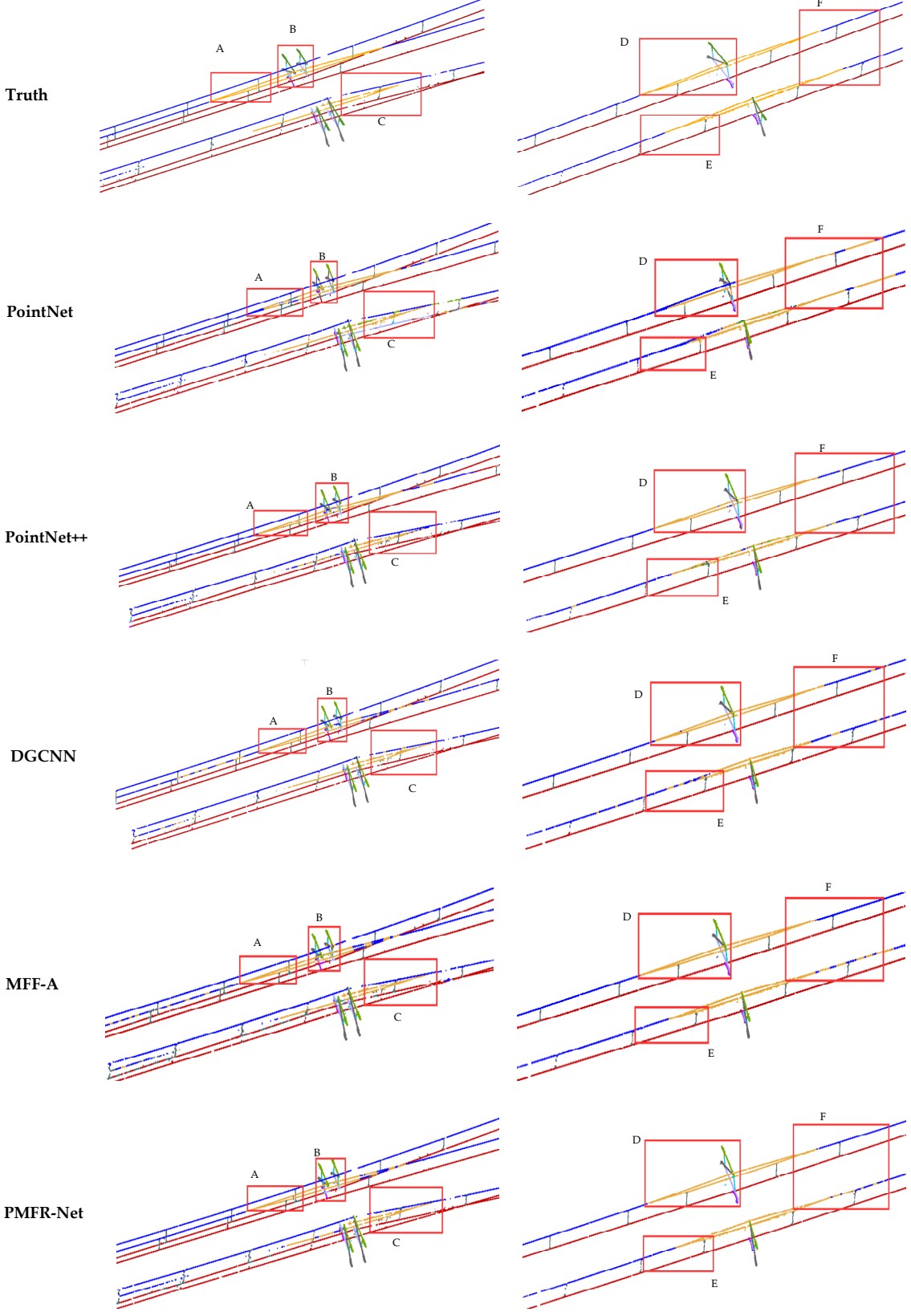

**Figure 9.** *Cont.*

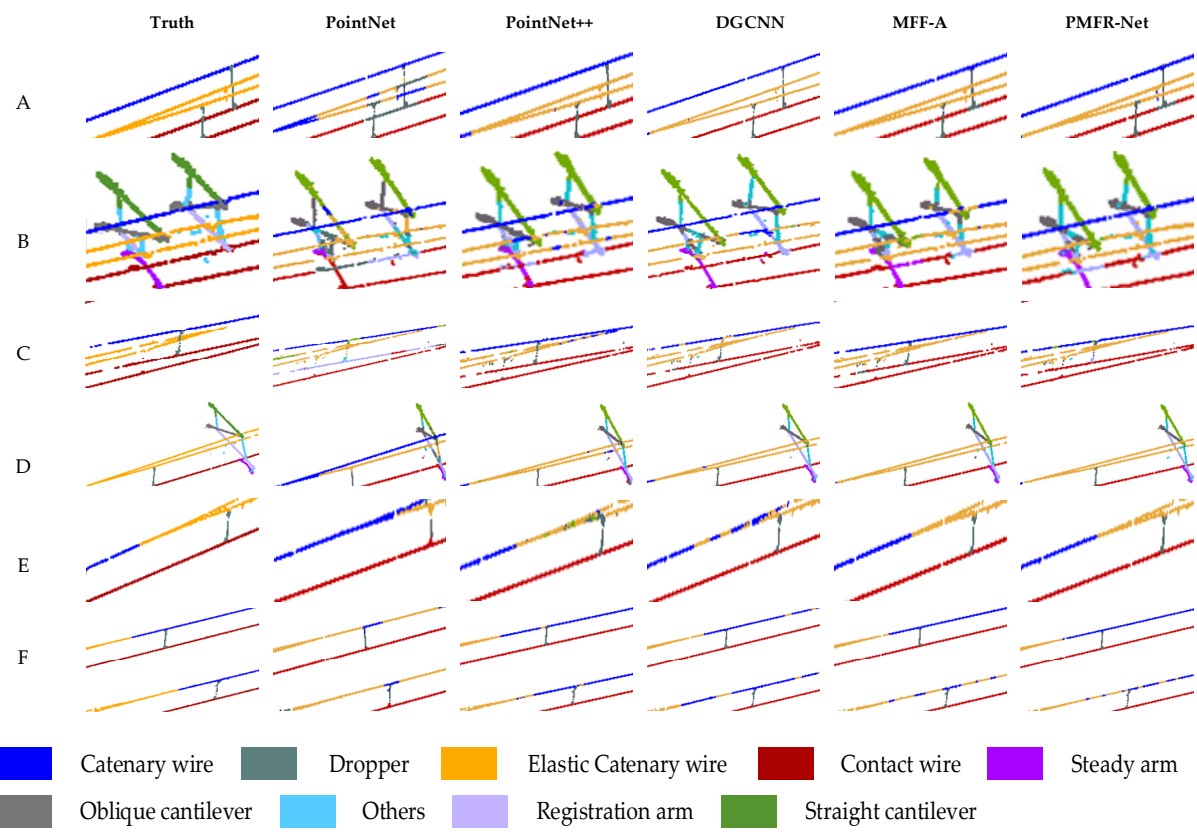

| | Catenary wire | | Dropper | | Elastic Catenary wire | | Contact wire | | Steady arm |
| | Oblique cantilever | | Others | | Registration arm | | Straight cantilever |

**Figure 9.** Semantic segmentation results of the OCS point cloud are based on different methods. The range delineated in the red box is enlarged to present a more detailed visual comparison at the bottom, where A, B, and C are the ROI in the double-arm scene, and D, E, and F are the ROI in the single-arm scene.

**Table 5.** Quantitative evaluation (%) of comparative methods on the S3DIS point cloud dataset. The best metric is highlighted in bold.

| Method | Precision | Recall | F1-Score | OA | MIoU | Time(h) |
|---|---|---|---|---|---|---|
| PointNet [24] | 83.29 | 81.33 | 80.69 | 92.36 | 70.69 | **7.5** |
| PointNet++(32) [25] | 90.59 | 91.56 | 90.78 | 95.45 | 79.69 | 33 |
| DGCNN [27] | 88.88 | 76.48 | 79.45 | 93.49 | 69.89 | 8 |
| MMF-A [26] | 93.99 | 92.76 | 93.28 | 96.65 | 87.96 | 9.5 |
| Ours | **96.42** | **94.94** | **95.63** | **97.72** | **91.84** | 8 |

## 4. Discussion

In this section, the ablation experiment explores the influence of the DECA module, SHDA, and PCRM structure on network performance. The ablation experiment in this section is divided into three parts. Section 4.1 studies the influence of the DECA module on network performance. Section 4.2 discusses the influence of SHDA on network performance. Section 4.3 examines the effectiveness of the PCRM structure on the network and discusses the dilation rate setting. All experiments in this study are based on a basic structure, PointSimpleNet, and the experimental data make up the OCS point cloud dataset introduced in Section 3.1. All experiments also conform to the experimental settings in Section 3.2.1 and will be evaluated using F1-Score, MIoU, and OA.

### 4.1. Ablation Experiment of Double Efficient Channel Attention Module

The original ECA module is compared with the DECA module in this paper. Three comparative networks with different ECA were designed. The first was the pri-

mary network, PointSimpleNet. The ECA and DECA are inserted behind the second and eighth convolution twice on the PointSimpleNet to construct the PointSimpleNet-ECA and the PointSimpleNet-DECA, respectively. The experimental results and semantic segmentation results of the partial OCS point cloud are shown in Table 6 and Figure 10. Results indicate that the attention mechanism helps improve the representation ability of the model. The DECA model proposed in this paper has a better effect than the ECA module, which proves the effectiveness of the DECA module.

**Table 6.** Quantitative evaluation of the DECA module through three groups of comparative experiments (%) on the OCS point cloud dataset. The best metrics are highlighted in bold.

| Method | F1-Score | MIoU | OA |
|---|---|---|---|
| PointSimpleNet | 89.89 | 82.91 | 93.66 |
| PointSimpleNet -ECA | 91.79 | 85.29 | 94.51 |
| PointSimpleNet -DECA | **92.13** | **85.68** | **95.22** |

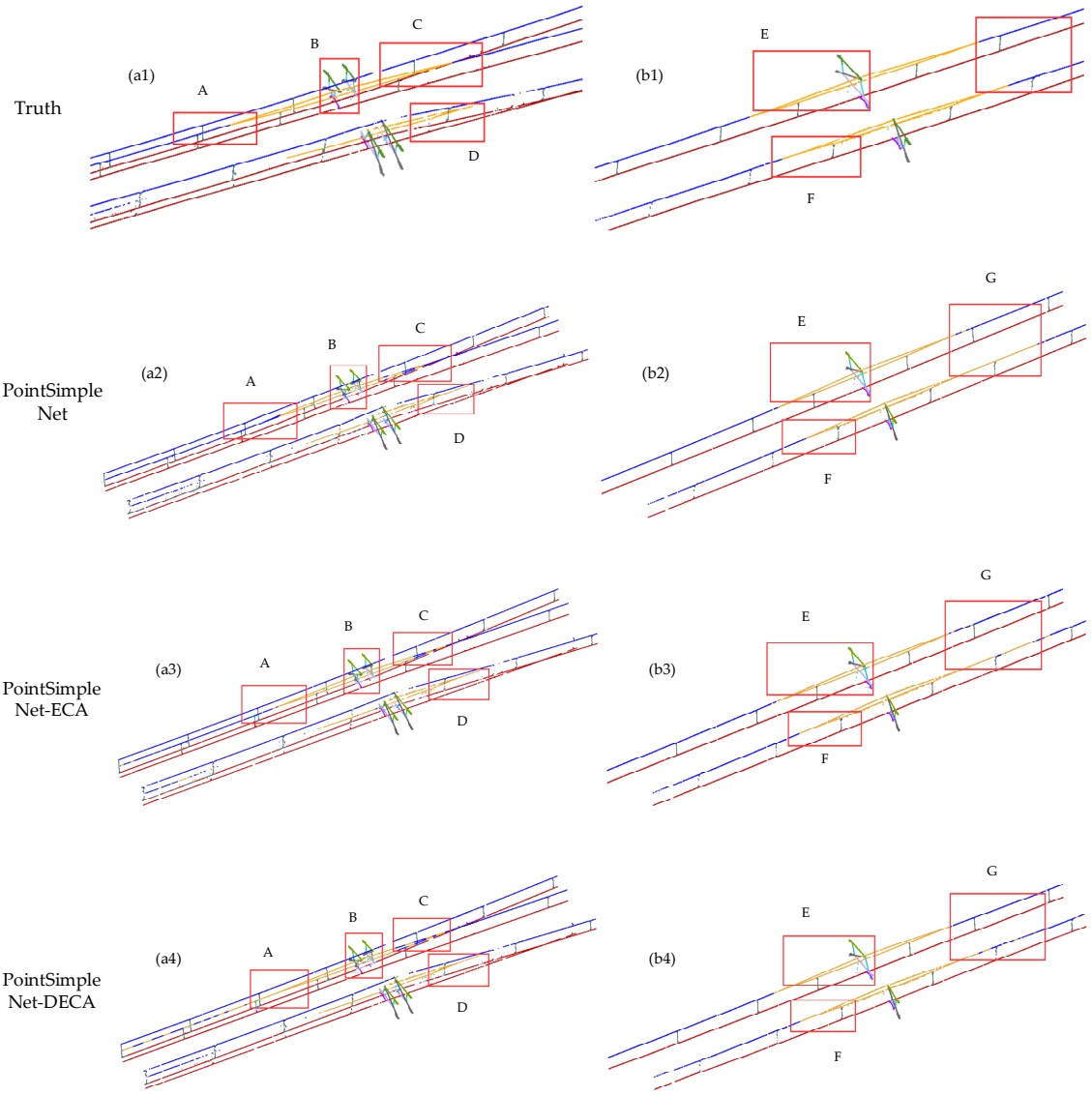

**Figure 10.** *Cont*.

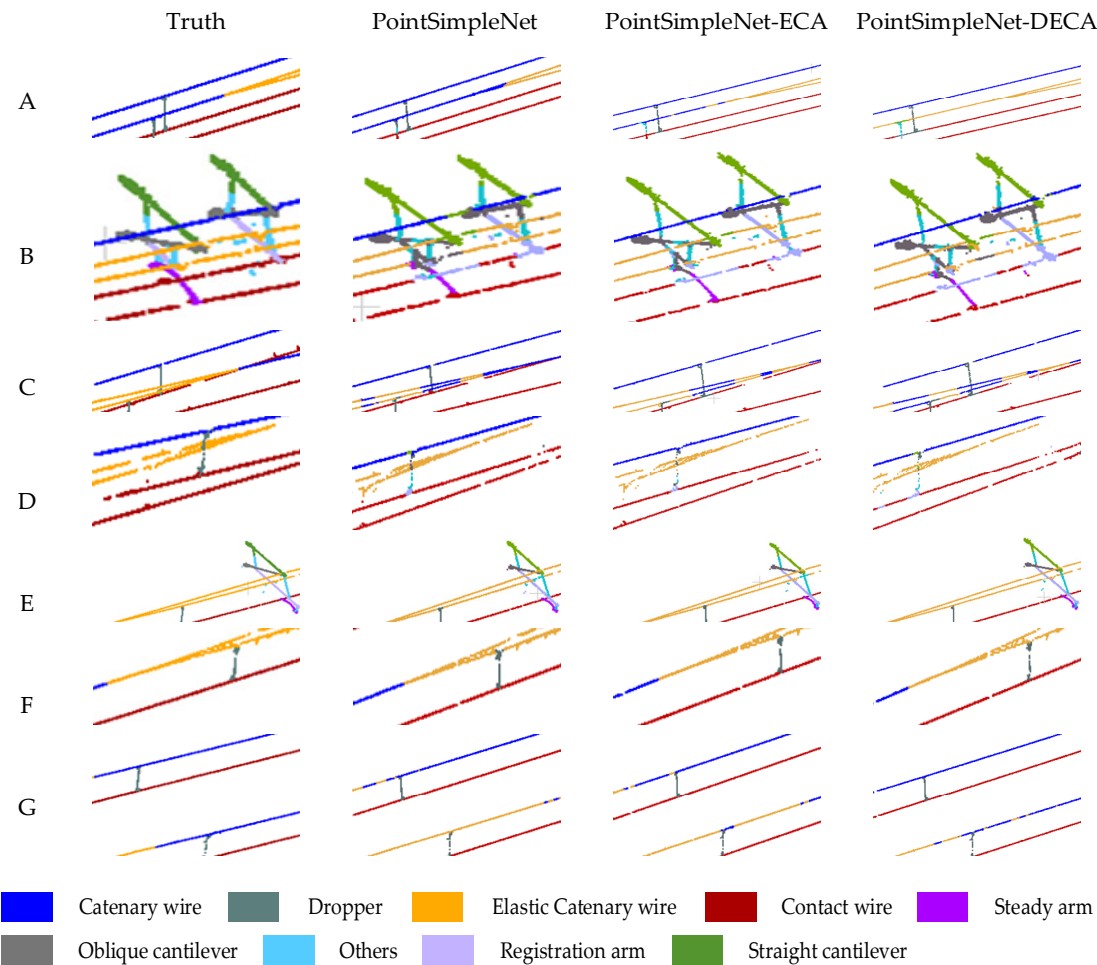

**Figure 10.** Comparison results of different attention mechanisms added to the basic frame. Column (**a1**–**a4**) and column (**b1**–**b4**) are global maps of double-arm and single-arm scenes, respectively. The range delineated in the red box is enlarged to present a more detailed visual comparison at the bottom, where A, B, C, and D are the ROI in the double-arm scene, and E, F, and G are the ROI in the single-arm scene.

By comparing the experimental results of PointSimpleNet, PointSimpleNet-ECA, and PointSimpleNet-DECA networks, the model representation ability of the DECA attention mechanism is the best. Compared with the basic framework, F1-Score, MIoU, and OA are improved by 2.24%, 2.77%, and 1.56%, respectively, since the added DECA module can filter redundant semantic information. As a result, the semantic information is more refined and easier to extract. As shown in region G in Figure 10, the boundary between the elastic catenary wire and the catenary wire in the single-arm scenes can be distinguished more accurately. There are still some misclassifications in the dropper, catenary wire, and elastic catenary wire. Compared to the basic framework, it has been significantly improved.

Compared with the original ECA, the DECA module is also improved to a certain extent. F1-Score, MIoU, and OA are improved by 0.24%, 0.39%, and 0.71%, respectively. Due to global max-pooling, certain information, such as boundaries, can be strengthened. Although the segmentation result of the DECA model still has certain misclassifications, as shown in Figure 9, it is superior to that of the original ECA model. Thus, it proves that parallel global max-pooling has certain effectiveness.

### 4.2. Ablation Experiment of the Serial Hybrid Domain Attention Structure

This paper uses the SHDA structure to connect contextual semantic information based on skip connections. Skip connections fuse the features of the encoding stage and decoding

stage. The comparison between the PointSimpleNet and PointSimpleNet-skip connections in Table 7 indicates that the skip connections can improve the accuracy to a certain degree. As shown in region C in Figure 11, the network with the added skip connections structure is more accurate than PointSimpleNet in processing the elastic sling boundary. From region A in Figure 11, it can also be found that the dropper part is misclassified after adding skip connections instead.

**Table 7.** Quantitative evaluation of SHDA structure through four groups of comparative experiments (%) on the OCS point cloud dataset. The best metrics are highlighted in bold.

| Methods | F1-Score | MIoU | OA |
|---|---|---|---|
| PointSimpleNet | 89.89 | 82.91 | 93.66 |
| PointSimpleNet-skip connections | 90.21 | 82.62 | 93.74 |
| PointSimpleNet-skip connections-ECA | 91.66 | 85.08 | 94.99 |
| PointSimpleNet-skip connections-SHDA | **92.03** | **85.42** | **95.12** |

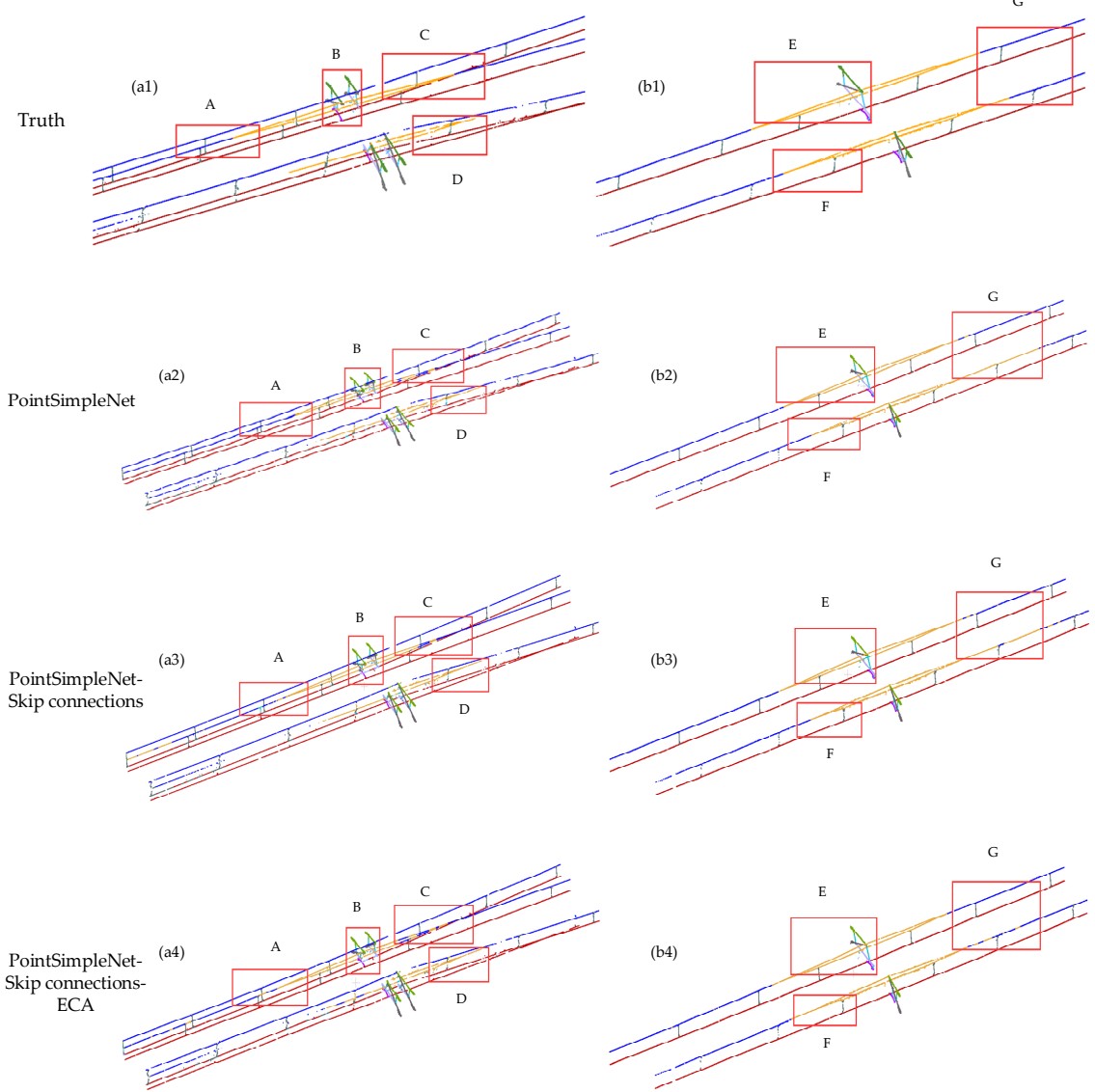

**Figure 11.** *Cont.*

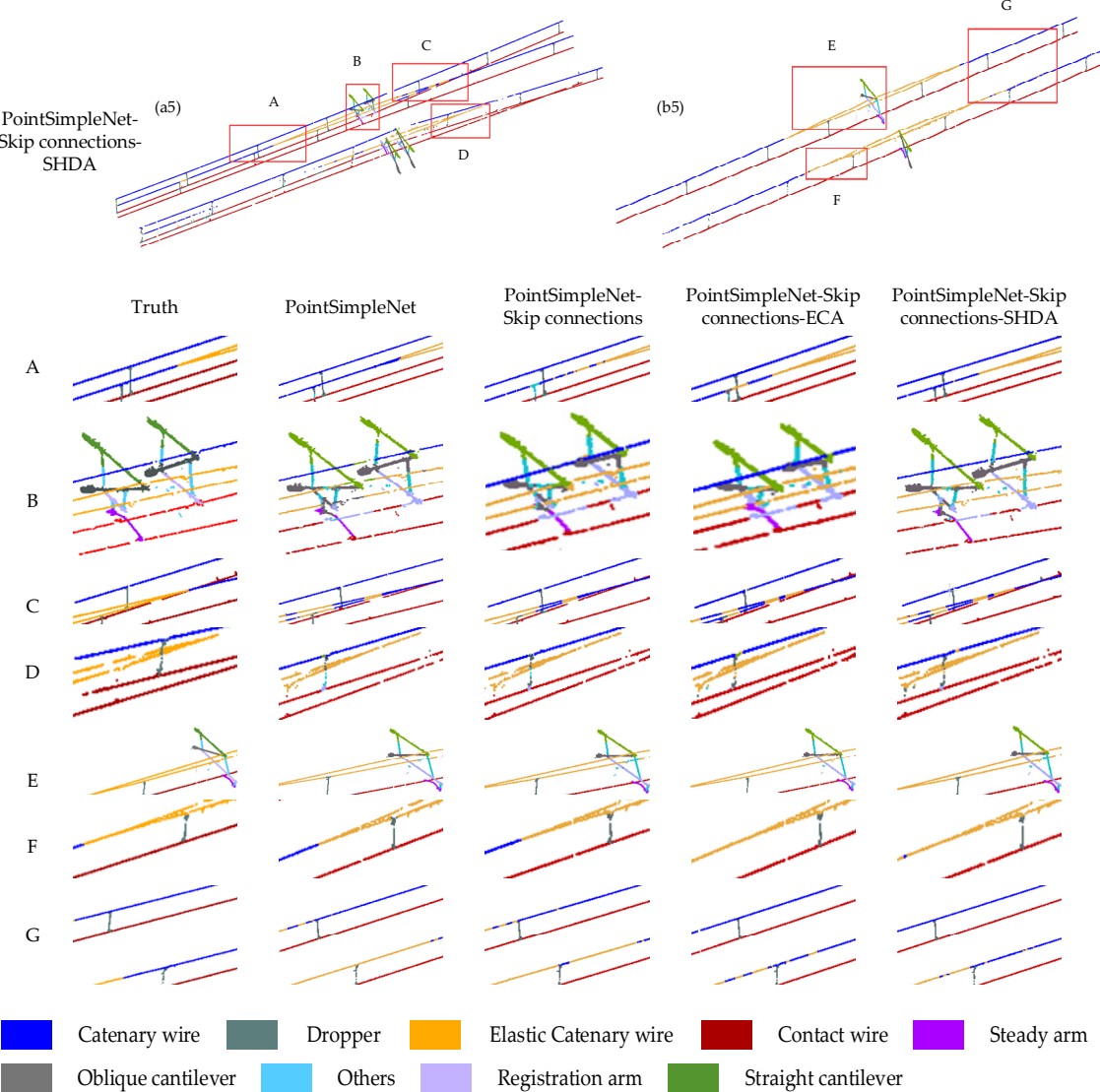

**Figure 11.** Comparison results between different connection methods added to the basic frame. Columns (**a1**–**a5**) and column (**b1**–**b5**) are global maps of the double-arm scene and the single-arm scene, respectively. The range delineated in the red box is enlarged to present a more detailed visual comparison at the bottom, where A, B, C, and D are the ROI in the double-arm scene, and E, F, and G are the ROI in the single-arm scene.

The SHDA can take the interaction between channels and the salient features in space simultaneously. This paper selects the channel attention mechanism and the skip connections structure as a comparison. Then, three connection structures are connected to the same level, respectively. The 4th and 8th layer convolution, the 5th and 7th layer convolution, and the 6th layer convolution and the max-pooling are fused, respectively. It can be seen from Table 7 that the serial attention mechanism is effective for improving the segmentation accuracy. Compared with only adding skip connection structures, the F1-Score of the serial ECA mechanism increased by 1.35% and the F1-Score of the SHDA mechanism increased by 1.82%. Compared with the channel attention structure (ECA is used in this paper), the SHDA structure has a 0.37% increase in F1-Score, a 0.34% increase in MIoU, and a 0.13% increase in OA. Therefore, the series mixed domain attention mechanism is superior to series channel attention. As shown in Figure 11, the SHDA structure effectively distinguishes the catenary wire and the elastic catenary wire. Thus, it is the most accurate and has a significant improvement.

### 4.3. Ablation Experiment of Point Cloud Refinement Module

The ablation experiment in this section is divided into two sections. Section 4.3.1 verifies the effectiveness of the PCRM proposed in this paper and Section 4.3.2 discusses the setting of dilation rate combinations in the refinement module.

### 4.3.1. Ablation Experiment of Point Cloud Refinement Module Structure

In this group of ablation experiments, three groups of comparative experiments are executed to prove the effectiveness of the PCRM. Based on PointSimapleNet, the ASPP [40] module and the PCRM are added as the refinement module to construct PointSimpleNet-ASPP and PointSimpleNet-PCRM, respectively. The ASPP module is a feature extraction module based on dilated convolution for the multi-level receptive domain perception of object features. The PCRM is a module that improves the feature fusion method based on the ASPP module. The experimental results show that both ASPP and PCRM improve the accuracy, since the refinement module strengthens the representation ability of the network.

As shown in Table 8, the PCRM has the highest semantic segmentation accuracy quantitatively. Compared with PointSimpleNet, the F1-Score, MIoU, and OA are increased by 1.25%, 2.97%, and 1.64%, respectively, proving that the PCRM can effectively optimize the results of PointSimpleNet and improve the segmentation accuracy of the whole network. As illustrated in Figure 12, the number of misclassified points was reduced using PCRM, especially in the connecting parts among components. For example, in the single-arm scene, many catenary wire points are identified as the elastic catenary wire points by PointSimpleNet. After adding the PCRM, nearly half of the points that were misclassified into elastic catenary wire were successfully identified as catenary wire, and the misclassification was corrected significantly.

**Table 8.** Quantitative evaluation of refinement module PCRM through three groups of comparative experiments (%) on the OCS point cloud dataset. The best metrics are highlighted in bold.

| Method | F1-Score | MIoU | OA |
| --- | --- | --- | --- |
| PointSimpleNet | 89.89 | 82.91 | 93.66 |
| PointSimpleNet-ASPP | 91.84 | 85.39 | 94.36 |
| PointSimpleNet-PCRM | **92.14** | **85.88** | **95.30** |

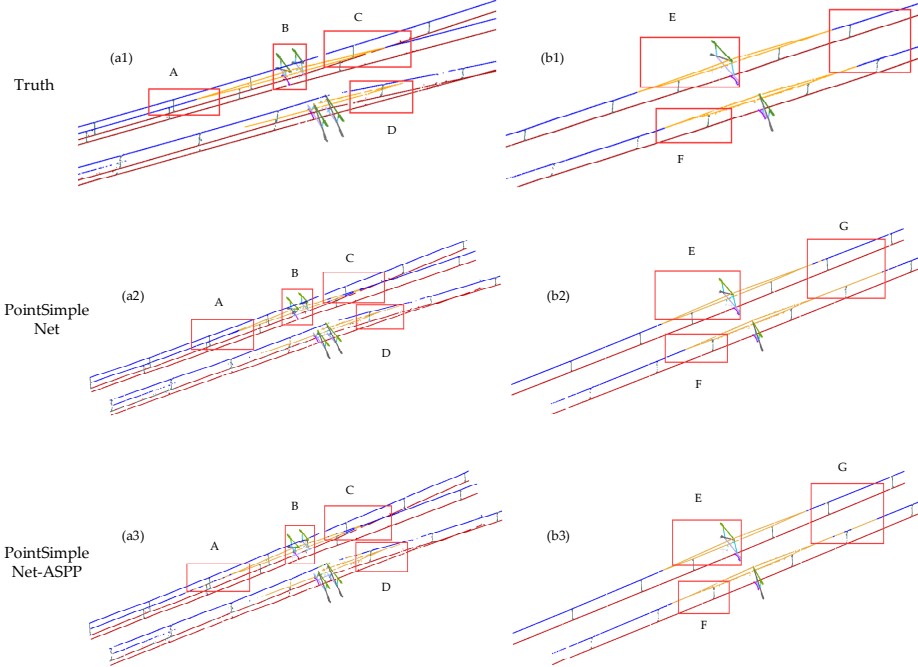

**Figure 12.** *Cont.*

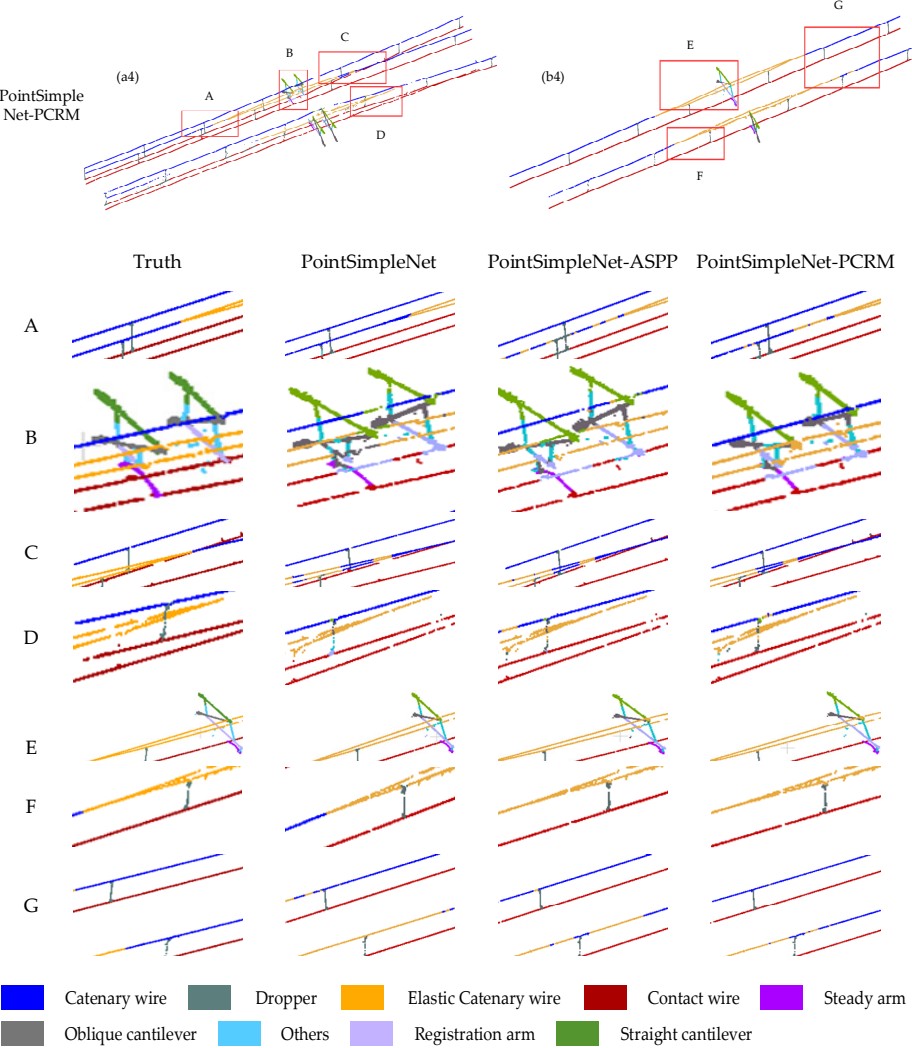

**Figure 12.** Comparison results of different refinement modules. Column (**a1–a4**) and column (**b1–b4**) are global images of double-armed and single-armed structures, respectively. The range delineated in the red box is enlarged to present a more detailed visual comparison at the bottom, where A, B, C, and D are the ROI in the double-arm scene, and E, F, and G are the ROI in the single-arm scene.

Compared with the ASPP, the PCRM improved F1-Score, MIoU, and OA by 0.3%, 0.49%, and 0.94%, respectively. The fusion method of multi-layer features is changed from simple unified fusion in ASPP to hierarchical sequential fusion in PCRM. The hierarchical sequential fusion can strengthen the information retention of several levels with a lower dilation rate and extract some details more accurately. Richer semantic information is better for semantic segmentation.

### 4.3.2. Discussion on the Dilation Rate Setting

This section mainly discusses the dilation rate in dilated convolution. The receptive fields of different sizes can be obtained by setting different dilation rates so that the characteristics of the same object at different scales can be obtained. Six comparative experiments were set up in this group. One was a reference experiment without dilated convolution. Based on the basic framework of PointSimpleNet, the optimal dilation rate combination is quantitatively analyzed by using different combinations of dilation rates. As shown in Figure 13, when the dilation rate combination is (1, 2, 4, 8, 16, 32), the three values of MIoU, F1-Score, and OA are the highest, reaching 85.88%, 92.14%, and 95.3%, respectively, and the three scores of the dilation rate combination reached a peak. That is because appropriate receptive fields can precisely contain training data and ensure

the integrity of extracted features. Excessive receptive fields would make some detailed features fuzzy and fail to provide accurate feature information. Therefore, the best dilation rate combination is (1, 2, 4, 8, 16, 32), which provides the most appropriate receptive fields.

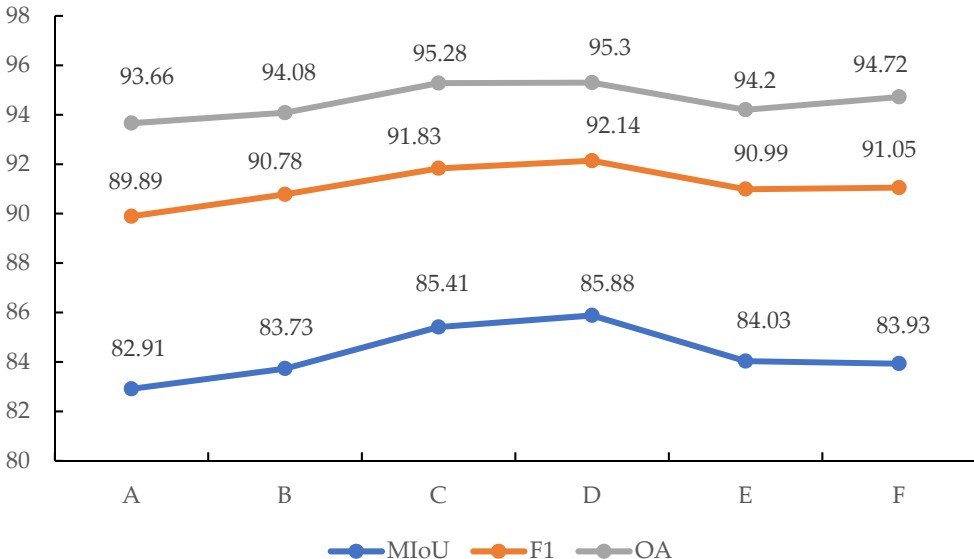

**Figure 13.** Line charts of three different precision evaluations for different dilation rate combinations, where (**A**) refers to experiment without dilated convolution, (**B**) refers to the combination of dilation rate (1, 2, 4, 8), (**C**) refers to the combination of dilation rate (1, 2, 4, 8, 16), (**D**) refers to the combination of dilation rate (1, 2, 4, 8, 16, 32), (**E**) refers to the combination of dilation rate (1, 2, 4, 8, 16, 32, 48), and (**F**) refers to the dilation rate combination of (1, 2, 4, 8, 16, 32, 48, 64).

## 5. Conclusions

In this paper, a deep learning network PMFR-Net was designed for semantic segmentation of the OCS point cloud, consisting of a prediction module and a refinement module. The prediction module inserted the DECA mechanism and the SHDA structure. The DECA was used to strengthen the extraction of local semantic features. The SHDA was used to filter cluttered semantic information of the features extracted in the encoding and decoding stage for better fusion. The DECA and SHDA can both strengthen the extraction of details to be adaptive for various OCS components. Then, the PCRM module formed the multi-scale feature extraction fusion module through multi-layer dilated convolution with different dilation rates. The original multi-level direct fusion mode was changed to multi-level sequential fusion, which can effectively reduce the loss of semantic information. The prediction module and refinement module complement each other and effectively improve the segmentation precision of the OCS point cloud. Experiments were carried out on the new OCS point cloud dataset proposed in this paper. The MIoU, OA, and F1-Score of PMFR-Net were 87.62%, 95.77%, and 93.24%, respectively, and the time was 4 hours. PMFR-Net is superior to SOTA methods in visual interpretation and quantitative evaluation compared with other comparison methods. Moreover, the S3DIS dataset is also introduced for the test, and the performance of PMFR-Net is better than the comparison methods. In addition, experiments were carried out on the setting of the dilation rate in the PCRM. In view of the degree of accuracy improvement, the optimal combination of dilation rates was (1, 2, 4, 8, 16, 32). MIoU and F1-Score could increase by 2.97% and 2.25%, respectively. At the same time, the effectiveness of the DECA mechanism, the SHDA structure, and the PCRM were validated by the ablation experiment.

In the future, we will improve PMFR-Net and pay more attention to the boundary processing between the components, especially the linear structure, to strengthen the processing of the details of catenary characteristics. At the same time, we will acquire more catenary point cloud data to expand the dataset and introduce data from different scenarios

of other railway lines in the dataset to train a more robust model and promote the practical application capability of the model.

**Author Contributions:** Conceptualization, T.X. and Y.Y.; methodology, T.X. and X.G.; writing—original draft preparation, T.X.; writing—review and editing, T.X., Y.Y., X.G. and Y.W.; supervision, Y.Y., L.X. and J.X.; funding acquisition, Y.Y. All authors have read and agreed to the published version of the manuscript.

**Funding:** This research is supported by the Open Fund of the National Engineering Laboratory for Digital Construction and Evaluation Technology of Urban Rail Transit (No. 2021ZH02); Open Fund of Hunan Provincial Key Laboratory of Geo-Information Engineering in Surveying, Mapping and Remote Sensing, Hunan University of Science and Technology (No. E22133, No. E22205); Open Fund of Beijing Key Laboratory of Urban Spatial Information Engineering (No. 20210205); and the Open Research Fund of the Key Laboratory of Earth Observation of Hainan Province (No. 2020LDE001).

**Data Availability Statement:** The OCS dataset was accessed from https://github.com/Waynexutao/DatasetAccess.git, (accessed on 4 May 2022).

**Conflicts of Interest:** The authors declare no conflict of interest.

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
