# Peer review of "Construction of a Semantic Segmentation Network for the Overhead Catenary System Point Cloud Based on Multi-Scale Feature Fusion"

_remotesensing, doi:10.3390/rs14122768_

Round 1

Reviewer 1 Report

the English language has been improved and the text is thus more understandable.

For convenience, the new revision notes have been made on the PDF submitted by the authors with comments on the revisions of the previous submission.

The dataset is still thought to be limited to serve as both network training and validation. In any case, in the notes it is suggested to insert a comment about it in the conclusions.

Furthermore, it is necessary to better specify which dataset the performance indices refer to.

Reviewer 2 Report

The paper proposes to use a deep neural network for performing semantic segmentation of overhead catenary point-cloud data. 

The proposed architecture for the neural network includes a prediction module and a refinement module. 

A data-set with catenary point-cloud is prepared and manually segmented for the experiments. A comparison against several architectures (Pointnet, Pointnet++, DGCNN, MFF-A) on the prepared data-set, as well as the S3DIS data-set, is provided.
An ablation study is used to show the impact of the different parts of the neural network architecture. 

The architecture of the neural network needs to be better described and be made more precise. 

Supplementary materials with precise details about the architecture such that the work can be reproduced should also be provided. 

The architecture proposed in this work looks very similar to the one proposed in [25]. While there are some modifications, they appear to be small improvements of existing ideas. The modifications are not always made very clear either. For example, it is difficult to understand the difference between ECA in [25] and DECA in this work (the equations are the same). 

The writing should be improved. There are several typos (some of them are provided below). 

Please find below more specific details about the manuscript. 

P. 1 line 36: Extra white space after ‘Still, ‘ (lack of white spaces and unnecessary white spaces appear in different places) 

Sometimes there is a space before introducing a reference (e.g. ‘edge-based [11]’ on line 55 p. 2), other times there is no space before the reference (e.g. ‘(SVM)[14]’ on line 63 p. 2). It is better to always use a white space before including a reference. 

P. 2 lines 60-61, the following sentence ‘Statistics-based methods are mainly machine learning’ is not clear. 

What is ‘Regular supervised machine learning’? 

Lines 71-78 on p. 2: The main problem with voxel data is that it is mostly useful for representing solid (or volumetric) models, while in the case of this work one is dealing with curves. The problem of resolution can be dealt with by using hierarchical structure. 

See, for example,
OctNet: Learning Deep 3D Representations at High Resolutions by Gernot Riegler, Ali Osman Ulusoy, Andreas Geiger
https://arxiv.org/abs/1611.05009

Line 83 on p. 2: What does ‘sed sampling’ mean?

Lines 88-89 on p. 2
Please check the grammar: ‘such as the catenary wire and the contact wire could not be detected effectively’ 

It would be better to introduce DGCNN [38] in Section 1 with the other methods for performing semantic segmentation on point-clouds. 

Line 119 on p. 3

Grammar: ‘are fused better’ 

Line 156: What are the ‘three shallow features’ fused through SHDA? Please provide better explanations. 

Lines 170-171 on p. 4 

Typo: ‘in a cross-channel manner.2.2.1 The Serial Hybrid Domain Attention structure’

Lines 178-179 on p. 4 

Typo: ‘Based on the CBAM[31].’ (This is not a complete sentence) 

In Figure 1, are the convolutions 1d? 

By ‘skip links’, do you mean ‘skip connections’? 

What is ‘f’ in equation (2)? 

Please indicate what equations (1) and (2) correspond to in Fig. 2. 

In equation (2), what does the notation ‘[A; B]’ correspond to? 

Line 196 on p. 5: ‘the nearest odd of t,’ -> The nearest odd number to t 

Line 197 on p. 5 ‘representd’ (Typo) 

Line 198 ‘k is the result’ -> ‘K’ (to use the same notation as in equation (3))

Line 198 on p. 5: What does ‘the adjacent channel of y_i’ correspond to? 

Which activation function is used for equation (4)? 

What are GAP and GMP in Fig. 3? 

On p. 6, lines 227-228, MF_i (i<n) is repeated two times: One time it corresponds to fused feature map, another time it corresponds to feature map. It seems to be a mistake. 

Please check the indices in equations (5) and (6). There are mixes of ‘i’ and ‘j’, which don’t look right. 

What is ‘convd’ in Figure 4? (The other figures use 'conv'. It is preferable to use the same term. Please also specify each time the dimension).

What is the point of writing ‘Accessed on 23 February 2022’ for your data-set? Does it mean that the data-set has been modified since then? 

In Table 1 (p. 8) why are the number of scenes in training and testing not matching the total number of scenes? (For example: number of D_arm scenes: 20, number of D_arm in training + number of D_arm in test = 19)

In Table 1, it would be better to use the full words rather than S_arm and D_arm. 

On p. 9, line 300: ‘All networking is implemented’ -> ‘All networks are implemented’

P. 9, line 302: ‘In our experiment, every point-cloud is represented by a 7D vector P = {X,Y,Z,R,G,B,I,T}’ -> P has 8 components, not 7. I assume that you meant that the last component T (the class) is not used in your experiment for the training. Please clarify.
What is the intensity corresponding to? Was it used for the training? In general, it is common to use XYZ only or XYZRGB. Is the intensity always going to be available? How much information does it carry? 

P. 9 line 305, it is HDF5 not H5. 

P. 9: ‘num_point’, ‘batch_size’ … Please avoid using variable names in the text (unless you include the corresponding code as well). 

Lines 373-385: PointNet and PointNet++ have already been introduced in the section on related works. I don’t think that it is necessary to repeat this information.
The explanations about DGCNN should go in the section on related works as well. 

It is difficult to distinguish the different results in Table 4. Please use lines to separate the results of the different approaches. 

In Fig. 8, the boxes used for the zoomed-in regions seem to be slightly different for each result. 

Line 488 ‘Figrue.9’ (Typo) 

Line 555 (p. 19) ‘the misclassification of points was relieved’. I don’t understand what you mean by ‘relieved’ in this context.

Line 602 on p. 22: There is a missing white space in ‘SHDAis’.

Just to confirm: In all the experiments the input is a point-cloud with size N*7, where N is the number of points, and each point has 7 components corresponding to: X,Y,Z,R,G,B,I? What kind of results are obtained if the intensity is not used? What happens if only the spatial coordinates are used (and not the color)?
Is it the case for the S3DIS data set as well (i.e. do you use X,Y,Z,R,G,B,I here as well)?
Were the methods that you compare to (pointnet++, DGCNN) modified to deal with such inputs (points are in R^7), and trained with such inputs as well?
The results provided in Table 5 are significantly better than the results documented in the literature. For example, DGCNN has a mean IoU of 69.89 while in [38] it is only 56.1. Why is it the case?

Round 2

Reviewer 2 Report

My opinion about this manuscript has not changed.

The writing has not really been improved. 

It is still not clear what the main new ideas behind the paper are (besides incremental improvements). 

There are still many problems with the notations. The architecture of the neural network is not made sufficiently precise (again, the authors can prepare supplementary materials with this information). 

P. 4: What is N in Nx512, Nx256, etc? 

In Fig. 1, the input is a 3D point cloud (a collection of points with their attributes), but the first arrow is a “conv2d Nx64”. How does it work (How is a 2D convolution, or rather a correlation, applied to a collection of points?) 

In Fig. 1, N is set to 4096. How are input point-clouds split and sub-sampled?

On p. 7 and 8, there are still the same problems with equations (6) and (7) as mentioned before.  

On p. 9 the data is 7D, while on p. 10 it is 8D. 

P. 10: What does it mean that the “num point is 4096”? What is a “num point”? 

P. 11: In (9), M represents the number of classes, while it is called n in Fig. 1.